# Wearable Wheelchair Mobility Performance Measurement in Basketball, Rugby, and Tennis: Lessons for Classification and Training

**DOI:** 10.3390/s20123518

**Published:** 2020-06-21

**Authors:** Rienk M. A. van der Slikke, Monique A. M. Berger, Daan J. J. Bregman, Dirkjan H. E. J. Veeger

**Affiliations:** 1The Hague University of Applied Sciences, 2501 EH The Hague, The Netherlands; m.a.m.berger@hhs.nl; 2Department of Biomechanical Engineering, Delft University of Technology, 2628 CN Delft, The Netherlands; D.J.J.Bregman@tudelft.nl (D.J.J.B.); H.E.J.Veeger@tudelft.nl (D.H.E.J.V.); 3Faculty of Behavioural and Human Movement Sciences, Vrije Universiteit Amsterdam, 1081 BT Amsterdam, The Netherlands

**Keywords:** wheelchair mobility performance, evidence-based classification, wearables, wheelchair sports

## Abstract

Athlete impairment level is an important factor in wheelchair mobility performance (WMP) in sports. Classification systems, aimed to compensate impairment level effects on performance, vary between sports. Improved understanding of resemblances and differences in WMP between sports could aid in optimizing the classification methodology. Furthermore, increased performance insight could be applied in training and wheelchair optimization. The wearable sensor-based wheelchair mobility performance monitor (WMPM) was used to measure WMP of wheelchair basketball, rugby and tennis athletes of (inter-)national level during match-play. As hypothesized, wheelchair basketball athletes show the highest average WMP levels and wheelchair rugby the lowest, whereas wheelchair tennis athletes range in between for most outcomes. Based on WMP profiles, wheelchair basketball requires the highest performance intensity, whereas in wheelchair tennis, maneuverability is the key performance factor. In wheelchair rugby, WMP levels show the highest variation comparable to the high variation in athletes’ impairment levels. These insights could be used to direct classification and training guidelines, with more emphasis on intensity for wheelchair basketball, focus on maneuverability for wheelchair tennis and impairment-level based training programs for wheelchair rugby. Wearable technology use seems a prerequisite for further development of wheelchair sports, on the sports level (classification) and on individual level (training and wheelchair configuration).

## 1. Introduction

Wheelchair basketball (WB), wheelchair tennis (WT) and wheelchair rugby (WR) are amongst the most popular Paralympic sports, with increasing popularity and international competitions being held worldwide. The rising level of professionalism and increased interests require more scientific performance insight, as a foundation to refine classification guidelines by the sport’s governing bodies and optimize training programs and conditions by the athletes and coaches.

In wheelchair sports, it is the interaction between athlete and chair that enables wheelchair propulsion and the movements across the court as required within a given sport [1]. Thus, in wheelchair performance, three components and their interactions need to be considered: the athlete, the wheelchair and the sport itself. Due to the large interindividual differences in athletes, insights based on able-bodied sports do not generally apply. Compared to able-bodied sports, wheelchair athletes face more challenges, while there is less scientifically based expertise available. As there is one common denominator, the use of a wheelchair, wheelchair mobility performance comparisons between sports might be a useful measure to increase evidence-based support.

In wheelchair court sports, overall game performance is highly dependent on individual wheelchair mobility performance [2]. These mobility performance outcomes regarding forward movement (speed and acceleration) and maneuverability (rotational speed and acceleration) typify the athlete and stipulate to what extent they can excel in a given sport, but also to what extent impairment type and level are important for sport-specific performance. The wheelchair mobility performance monitor (WMPM) [2] was used to quantify the differences in WMP during match-play between wheelchair tennis, wheelchair rugby and wheelchair basketball. This measurement method is the first to provide objective and well quantified wheelchair performance data, in actual sport settings (training session, match-play) and across different sports. It proved valid and reliable in different sport settings [3,4,5].

Wheelchair mobility performance closely relates to the athlete–wheelchair interaction, covering two of the main components in wheeled sports (athlete and wheelchair) [6,7,8]. By measuring across sports, also the third performance component (sport) is addressed, providing insight in the resemblances and differences across sports. For designing training protocols, it is interesting to know which wheelchair mobility performance aspects are sport- or impairment-level specific. If discrepancies and similarities are known, it becomes easier to exchange knowledge across sports. Can a sprint training protocol for low-class wheelchair basketball athletes be used for wheelchair rugby players as well? Can an agility training scheme be used across all wheelchair rugby players, or does it require fine tuning per class? Regarding the classification rules and regulations, insight in required wheelchair mobility performance levels per sport could provide a better base for refining the different classes. Are all wheelchair mobility performance aspects equally important for all sports? Do athletes with similar impairment levels show equal mobility performance levels in the different sports? Can lessons be learned for classification, training and wheelchair setup if more WMP insight at an elite level of match-play across sports is available?

The three wheelchair sports have different eligibility rules and use dissimilar ways to assess function level for classification. The International Paralympic Committee described the eligibility criteria for impairment type for all Paralympic sports [9], but the severity of impairment (impairment level) for eligibility differ among sports. If eligible, athletes are classified according to the extent of activity limitation caused by the impairment level, and its effect on sport-specific performance. To ensure fair competition, a selective classification system is required, so athletes who enhance their competitive performance through effective training will not be moved to a class with athletes who have less activity limitation [9]. Yet, such a classification system in turn requires insight in the relationship between impairment levels and the sport-specific on-court performance. Since at present only limited scientific evidence is available to describe those relationships for each sport, classification methods tend to be more or less hybrid, with off-court function (impairment level) assessment, as well as on-court function assessment by observation. Yet, on-court function assessment tends to be subjective and is easily influenced by the athlete’s performance levels. Wearable objective wheelchair mobility performance measurements, that can be used across sports and both during match-play and isolated field tests, could aid in refining the classification system as well as align them across wheelchair court sports.

For wheelchair tennis, all athletes are eligible (in the open class), but as a consequence only athletes with the least impairment towards upper extremity survive in competition. The International Tennis Federation however, considers introducing more classes than the current open and quad class. In wheelchair basketball, impairment levels range from minor (e.g., amputation of the first ray of one foot) to severe (e.g., C5 SCI), whereas in wheelchair rugby, only athletes with irreversible upper extremity impairments are eligible (typically athletes with tetraplegia), so with an even more severe effect on WMP. In wheelchair rugby, function is assessed with isolated tests, with assumptions about their effect on game performance. In wheelchair basketball, function is assessed by match observation, which is ecologically valid, but given the match specific factors, it is not always the athlete impairment level that determines observed performance [10]. Comparing the different sports could uncover in which sports and performance aspects, WMP is the limiting factor in game performance. Regarding training and wheelchair setup, knowledge is often only shared within a specific sport since there are many sport-specific characteristics. Considering wheelchair properties, weight and weight distribution are one of the most discriminative factors between the three sports, mainly determined by whether or not contact with opponents is allowed within a given sport. Wheelchair tennis (no contact) has the lightest wheelchairs, wheelchair basketball more robust and heavier ones, and wheelchair rugby the most bulky and heavy chairs (see Section A.1). The sport itself also comes with specific properties affecting WMP, such as the court dimensions, ball and racket handling, number of players and team setup, and the intensity and duration of a match (see Section A.2). Yet, if WMP across sports where participating athletes have similar impairment levels shows much resemblance, that might give cause to improve knowledge exchange among sports rather than only addressing the differences.

For wheelchair basketball, wheelchair tennis (open class) and wheelchair rugby, this research will describe if WMP is a critical factor in the specific sports performance; what WMP aspects per sport are most important; what WMP outcomes should shape classification guidelines and in what performance areas knowledge could be exchanged between sports. Furthermore, it stipulates the added value of wearable technology use to advance wheelchair sports, both on the sports level (evidence-based classification guidelines) as well as on an individual level (training and wheelchair configuration).

## 2. Materials and Methods

### 2.1. Participants

Wheelchair mobility performance was measured in match-play [2] for elite level athletes in wheelchair basketball athletes (*n* = 29), wheelchair rugby players (*n* = 32) and wheelchair tennis players (*n* = 15, no quads, Table 1). The wheelchair basketball athletes were measured during eleven premier division competition matches and friendly international level matches (GBR, NLD, ISR and AUS). The wheelchair rugby players were measured during the Dutch national championship of 2016, during a practice match of the Dutch national team and at the Amsterdam Quad Rugby Tournament 2017 (NLD, CHE, DEU, FRA, BEL, CZE). Measurements for athletes with substitutions were only included if they were playing for at least half of the match. Thus, over 16 min active game time for wheelchair rugby and over 20 min for wheelchair basketball. Finally, wheelchair tennis players of the open class were measured during the 2016 Dutch championship and the international ABN-AMRO wheelchair tennis tournament of 2017 (NLD, ARG, FRA, GBR, RSA, ESP). The pathologies for the wheelchair tennis included spina bifida, SCI (L4–5), congenital reduction defect syndrome, Perthes disease and (partial) leg amputations, with all impairments only significantly affecting lower extremity function.

For wheelchair basketball and wheelchair rugby, this study was approved by the ethical committee of the department of Human Movement Sciences, Vrije Universiteit Amsterdam, and for wheelchair tennis measurements by the Human Research Ethics Committee of the TU Delft. All participants signed an informed consent form after being informed on the aims and procedures of the experiment.

### 2.2. Methodology

Each athlete’s own sports wheelchair was equipped with three inertial sensors (x-IMU for wheelchair basketball, X-IO technologies, x-io.co.uk; Shimmer3 for wheelchair tennis and rugby, Shimmer Sensing, shimmersensing.com), one on each rear wheel axis and one on the rear frame bar, as described by van der Slikke et al. [3,4] (Figure 1). The frame sensor provides forward acceleration data as well as rotation of the frame in the horizontal plane. The combined signal of wheel sensor acceleration and gyroscope was used to estimate wheel rotation, which in turn provided frame displacement given the wheel circumference. Frame sensor gyroscope data were used to correct the wheel gyroscope signal for wheel camber angle, as described by Pansiot et al. [11], Fuss et al. [12] and van der Slikke et al. [3]. Furthermore, a skid correction algorithm was applied to reduce the effect of single or concurrent wheel skidding [4].

This WMPM measurement setup [2] provides a standardized plot with six key kinematic performance outcomes. In the WMPM analysis, each measurement is divided into sections of speed >0.1 m/s and rotational speed (change of heading direction) >10 °/s, on which the following outcomes are calculated: average speed; average best speed (of best 5 runs/speed sections); average acceleration in the first 2 m from standstill; average rotational speed during a curve (forward speed above average); average best rotational speed during a turn on the spot (of best 5 turns/rotational speed sections, with forward speed below average speed) and average rotational acceleration. In this method, the cut off speed between “turn” and “curve” is the average forward speed, previously determined as 1.5 m/s in wheelchair basketball [2], a speed of 1.3 m/s for wheelchair tennis and a speed of 1.1 m/s for wheelchair rugby.

In addition to the standard WMPM outcomes, the distribution of time spent in certain speed zones is calculated, as described for wheelchair rugby [13], wheelchair tennis [14] and wheelchair basketball [15]. Based on the used thresholds, a division was made in 0–0.5 m/s; 0.5–1.5 m/s; 1.5–2.5 m/s and over 2.5 m/s, and additionally negative speeds were included as a separate speed zone. The WMPM also measures rotational speeds, so time spent in rotational speed zones could also be calculated [16]. Absolute (disregarding left or right turning) rotational speeds were classified in 0–25 °/s; 25–50 °/s; 50–100 °/s; and 100+ °/s.

To estimate to what extent average impairment level per sport influenced performance outcomes, a separate analysis was made for the least impaired athletes only. This separate selection was added, since for wheelchair tennis only the least impaired athletes were included (solely athletes from the open class participated). For the other two sports, this “high-class” group consisted of class 2.5+ athletes for wheelchair rugby, of class 3+ athletes for wheelchair basketball. These athletes are at least more comparable since athlete population per sport does not allow for complete impairment level matching across sports.

For all outcomes, means, standard deviations and differences between sports were calculated, both for the entire group as well as for the “high-class” selection. The significance of the difference was determined using a one-way ANOVA with Bonferroni post hoc test (*p* < 0.05), preceded by a Kolmogorov–Smirnov test for normal distribution.

## 3. Results

Average speed was highest in wheelchair basketball (WB, 1.57 ± 0.13 m/s), followed by wheelchair tennis (WT, 1.34 ± 0.13 m/s) and wheelchair rugby (WR, 1.13 ± 0.27 m/s). A similar order was visible in the maximal speeds achieved (WB = 4.98 ± 0.43 m/s; WT = 4.40 ± 0.40 m/s and WR = 3.37 ± 0.99 m/s) and in the maximal rotational speeds (WB = 388 ± 71 °/s; WT = 369 ± 79 °/s and WR = 303 ± 43 °/s). For the WMP outcomes (Figure 2), the order of sports is the same, with highest performance values in WB, closely followed by WT and WR last. Only in rotational speeds WT athletes slightly outperform WB athletes. Within the selection of “high-class” players, the differences between WR and WT decrease (Figure 3). Table 2 shows all mean values and standard deviations, whereas an overview of significant differences between sports are included in the Appendix A.

In all sports, a substantial amount of time, ~10% was spent in reversed speed (Figure 4). Most of the time was spent in the zone that incorporated the average speeds of tennis and rugby (0.5–1.5 m/s). In wheelchair tennis, the most time above average speed was spent in the 1.5–2.5 m/s zone, with only minimal time in the 2.5+ m/s zone. For wheelchair basketball, also considerable time was spent in the 2.5+ m/s speed zone.

In rotational speeds, differences between distribution in zones were less prominent, despite the fact that some differences between sports were still quite statistically significant (*p* < 0.05; Figure 5). The one clear deviation was the additional time of wheelchair rugby players in the lowest zone and the reduced time in the 100+ °/s zone.

## 4. Discussion

### 4.1. Wheelchair Mobility Performance Quantified

Wheelchair basketball showed the highest wheelchair mobility performance outcomes, and the most time spent in high (rotational) speed zones. As expected, wheelchair tennis mobility performance followed, with rotational speeds and time spent in high rotational speed zones quite similar to basketball. Wheelchair rugby players show the lowest mobility performance, which is most likely based on the use of heavier wheelchairs and highest level of impairment of the athletes. Five out of six WMP outcomes were significantly lower for wheelchair rugby compared to both wheelchair basketball and wheelchair tennis (Appendix A, Table A1). With only the least impaired wheelchair rugby athletes selected, the differences with wheelchair tennis in all WMPM outcomes diminished, only the average starting acceleration still showing significant differences. Thus, even though the athletes in the “high class” wheelchair rugby group were on average still more impaired then the wheelchair tennis athletes, only minimal performance differences showed. The WMP differences between the least impaired wheelchair rugby and basketball athletes maintained, so it is likely that the initial differences found in wheelchair tennis were mainly a consequence of the lower classified athletes in wheelchair rugby. This finding is in line with van der Slikke et al. [2], who concluded that there was a steep wheelchair mobility performance drop in athletes with low classification (class < 2) compared to the remaining athletes (class 2+). In a study comparing different divisions in elite wheelchair tennis players, Mason et al. [16] concluded that most performance characteristics are significantly lower for the quad class, compared to the open class athletes. Given those results, it is assumable that the quad class players will also perform less than the “high-class” wheelchair rugby players. How their performance compares to “low-class” players could be part of future research.

Even with the “on average” higher impairment level of athletes in wheelchair basketball compared to wheelchair tennis, four out of six performance outcomes show higher values; only the rotational speed ranges are similar. The absence of a nearby opponent reduces the need for abrupt maneuvers, so motions will be more fluent and less intense (less forward or rotational accelerations). Court size differences could explain why forward speeds are lower compared to basketball, whereas rotational speeds are not.

The wheelchair mobility performance profiles represent match-play, so with all sport-specific influences taken into account, a more isolated field test could be used to determine the athlete’s maximal wheelchair mobility performance as described by van der Slikke et al. [17] and Rietveld et al. [5]. In such a test, wheelchair tennis athletes will likely range higher, as they are on average the least impaired. Those additional field test measurements could discriminate if differences in mobility performance level could be attributed to sport-specific limitations (e.g., court size, opponents) or to individual athlete performance level, an important factor for consideration in the chosen classification method. If determination of impairment level is partially based on match observation, this might misrepresent the maximal performance level of the athlete.

For the quantification of the differences presented, it should be kept in mind that this study has some limitations. It is difficult to perfectly match athletes of the different sports measured, for competition level, match time, and so on. The athletes included were all elite level athletes and matched as well as possible for competition level and sex, but different athlete selections could have provided slightly different results. Once more performance data of athletes in competition are present, selections for comparison could be better matched for competition level, impairment level (classification) and sex. Another aspect that needs to be regarded, is that the WMPM was initially developed for wheelchair basketball [2], with a selection of key wheelchair kinematic outcomes based on basketball match-play. To assure that the same kinematics typify the other sports as well, a similar procedure to extract key outcomes should be executed.

### 4.2. Implications for Classification and Training

Wheelchair basketball shows the highest mobility performance, with the highest average speed. Considering WMP, it is the most demanding sport of the three, so for classification, these results could be employed to incorporate endurance and intensity aspects of performance. To that extent, the current WB classification system with match-based assessment of function does address these aspects. Regarding training, both forward motions, as well as maneuverability exercises should be incorporated at high intensity (relative to WT and WR), where (rotational) acceleration seems of more importance than achieving maximal speeds. Regarding wheelchair configuration, weight and weight distribution should be optimized for acceleration and agility [7,18], within the restrictions of a contact sport. In wheelchair tennis, WMP does not seem to be the most critical factor in game performance, since WMP levels show lower than the even more impaired peers in wheelchair basketball. Therefore, regarding the classification, the effect of impairment level on WMP should perhaps weigh less than the effect of impairment level on other performance aspects. Although, part of the lower WMP could also be explained by the effect of racket use on propulsion, resulting in lower (rotational) accelerations, as also previously described by de Groot et al. [19]. The outcomes shown could be enforced to optimize training and wheelchair configuration for less intense movement, or it could be argued that a counter action is required to enhance performance. Currently, tennis strokes are made while rolling, whereas potentially stroke precision could increase when speed is reduced, therefore, requiring the need to accelerate in between strokes and decelerate before a stroke. In wheelchair rugby, WMP outcomes show the highest variation, most likely due to higher impairment level variation within this sport. The large performance range emphasizes the need for a fair classification system, preferably based on objective measures of wheelchair mobility performance measurement and its relationship with sports performance. Regarding training, the large performance differences within the rugby athletes pinpoint the need for a differentiated training program. High point players could do with training schedules, similar to wheelchair basketball, but low-point players might benefit more from training at lower (rotational) speeds that better fit their functional capacities. In rugby, the requirement for a solid wheelchair that can withstand impact is leading in the design and configuration. Nevertheless, given the overall weight due to the firm frame, weight distribution could still be optimized for maneuverability.

In prior research, reversed speeds were often neglected, but the WMPM revealed that athletes drive backward about 10% of the time in all sports. Therefore, this activity should be incorporated in the function assessment for classification and preferably as part of the training routine.

### 4.3. WMPM Use in Sports Practice

Apart from the group level recommendations, the wheelchair mobility performance monitor also allows for individual monitoring of each athlete. This is particularly beneficial in wheelchair sports, given the heterogeneity of athletes, with very individual differences in capacities and impairment levels. Individual monitoring could be used to optimize wheelchair configuration, evaluate training programs and watch the course of a match season. The use of this wearable technology empowers the athletes and coaches to evaluate their training and match performance in a quantitative and objective manner, and act upon it. High quality objective performance monitoring is key to match preparation, not only to enhance performance, but also to reduce overuse injury risk. Cooper at al. [20] described the communalities and differences between wheelchairs for the different sports, as well as stretches needed for optimal wheelchair configuration, not only to attain the best possible performance but also to reduce the risk of injuries.

## 5. Conclusions

The wheelchair mobility performance monitor showed performance profiles per court sport in line with the expectations, with wheelchair basketball showing highest mobility performance and wheelchair rugby the lowest. The WMP profiles could be used to refine classification systems and training programs since it pinpoints to what extent WMP is important to the sport-specific performance. In wheelchair tennis, WMP does not seem the most critical factor, since assumedly no maximal WMP levels are attained in match-play, whereas in wheelchair rugby, the huge WMP level variation indicates its importance for sport performance. In a new version of the WMPM, trunk movement and heartrate are also included in the analysis, allowing for better estimates of performance intensity and the effect of trunk impairment level on performance. Both key aspects in the classification of all sports are included [21].

Apart from the proven sensitivity for quantifying performance differences between competition and impairment level [10,16,18,22], this study also shows its applicability across wheelchair court sports. The WMPM could support individual athletes in performance enhancement within a match or within a season, it could support coaches in monitoring players and selecting the best team composition, it could support wheelchair specialist in tuning the wheelchair configuration for best possible performance and, finally, it could aid sport governing bodies in optimizing classification guidelines.

## Figures and Tables

**Figure 1 sensors-20-03518-f001:**
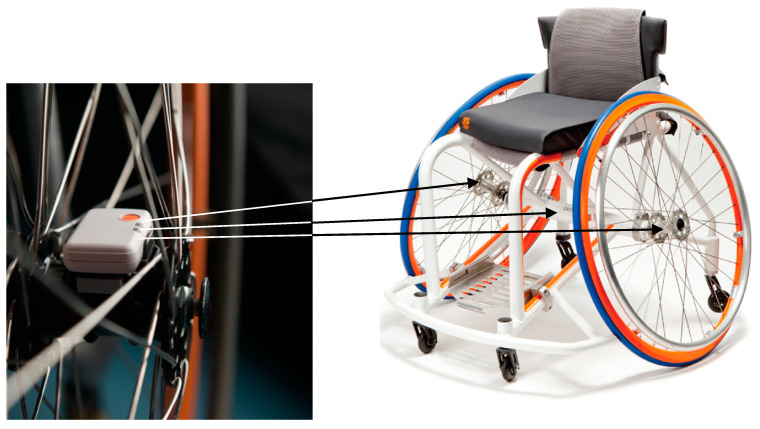
Sensor placement on the wheelchair, with one on each wheel axle and one on the frame camber bar (sensor picture by Marc Hollander Photography: www.marchollander.nl).

**Figure 2 sensors-20-03518-f002:**
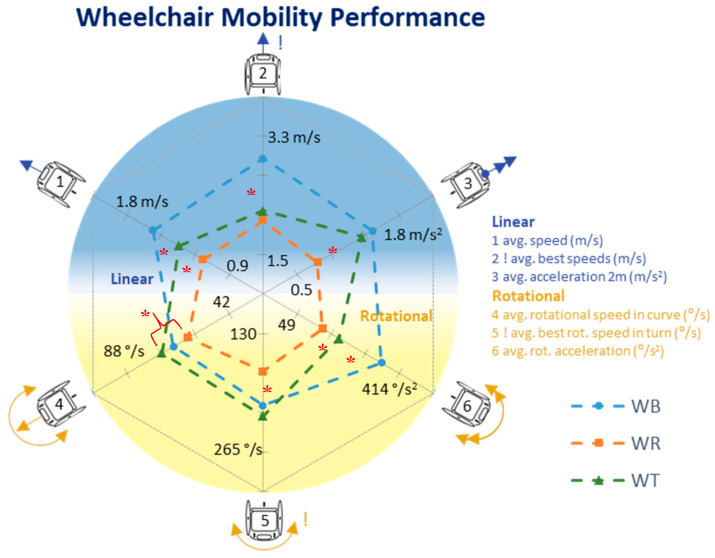
WMP for all sports. Significant (*p* < 0.05) differences are marked with a *. See Table A1 for the values (Appendix A).

**Figure 3 sensors-20-03518-f003:**
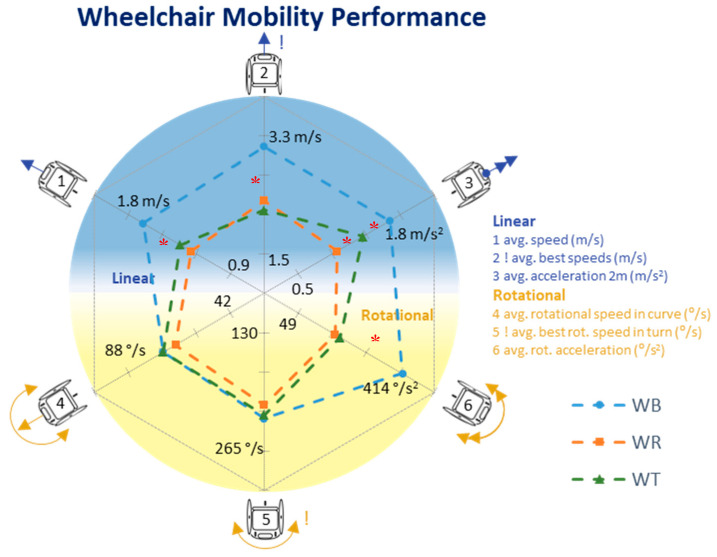
WMP for all sports for “high-class” athletes only. Significant (*p* < 0.05) differences are marked with a *. See Table A1 for the values (Appendix A).

**Figure 4 sensors-20-03518-f004:**
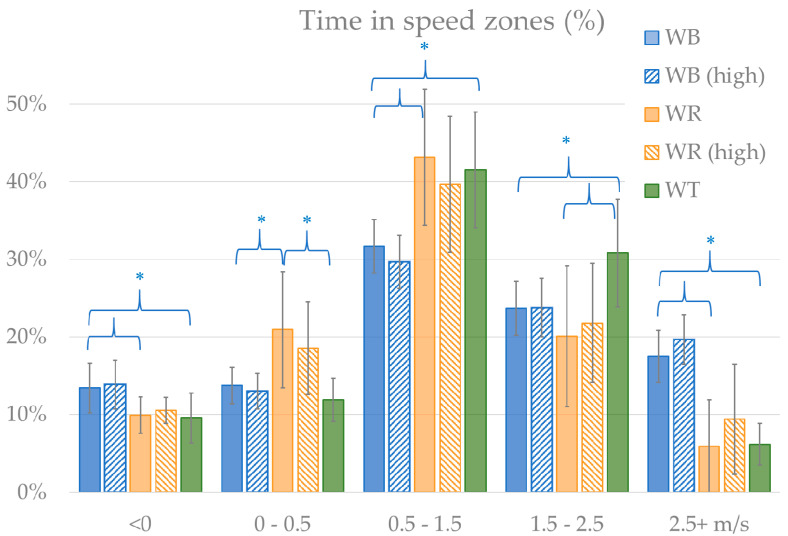
Distribution of time spent in speed zones, with significant (*p* < 0.05) differences marked with *.

**Figure 5 sensors-20-03518-f005:**
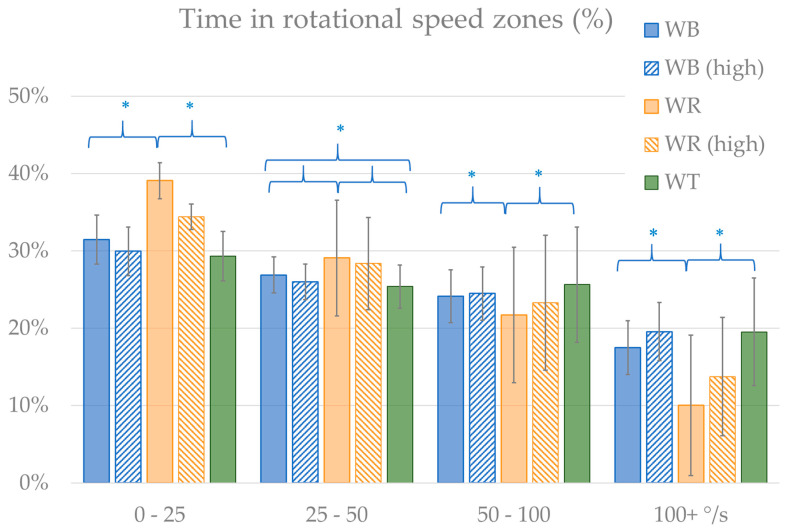
Distribution of time spent in rotational speed zones, with significant (*p* < 0.05) differences marked (*).

**Table 1 sensors-20-03518-t001:** Participant characteristics. For the class, additional grouping was applied with a low-class group consisting of class 1–2.5 for wheelchair basketball and 0.5–2 for wheelchair rugby. The high-class group consisted of 3+ for wheelchair basketball, 2.5+ for wheelchair rugby and all athletes for wheelchair tennis.

Sport	Class											Class	Sex	Level
	Total	0.5	1	1.5	2	2.5	3	3.5	4	4.5	NA	Low/High	M/F	Int./Nat.
Basketball	29		5	4	3	5	3		4	5		17/12	20/9	12/17
Rugby	32	5	4	3	8	4	5	3				20/12	29/3	13/19
Tennis	15										15	0/15	4/11	5/10

**Table 2 sensors-20-03518-t002:** Averages and standard deviations for the wheelchair mobility performance outcomes, the maximal values and the time spent in (rotational) speed zones for the three sports. The upper table shows the overall outcomes and the bottom table only the “high-class” athletes (WB, 3+; WR, 2.5+; WT, all).

	**All**	**Total (n = 76)**		**Basketball (n = 29)**		**Rugby (n = 32)**		**Tennis (n = 15)**	
		**Mean**	**Stdev.**	**Mean**	**Stdev.**	**Mean**	**Stdev.**	**Mean**	**Stdev.**
**WMP**	Avg. forward speed (m/s)	1.34	0.28	1.57	0.13	1.13	0.27	1.34	0.13
	Best forward speed (m/s)	2.41	0.60	2.96	0.23	2.02	0.59	2.17	0.14
	Avg. acceleration to 2 m (m/s^2^)	1.14	0.45	1.46	0.26	0.76	0.35	1.33	0.29
	Avg. rotational speed in a curve (°/s)	65	15	67	9	60	21	72	8
	Best rotational speed in a turn (°/s)	198	45	212	28	173	52	224	29
	Avg. rotational acceleration (°/s^2^)	232	122	351	110	139	44	199	26
**Max**	Maximal forward speed (m/s)	4.19	1.02	4.98	0.43	3.37	0.99	4.40	0.40
	Maximal rotational speed (°/s)	348	80	388	71	303	79	369	43
**Speed zones**	Speed <0 m/s	11.2%	3.3%	13.4%	3.2%	9.9%	2.3%	9.6%	3.2%
	Speed between 0 to 0.5 m/s	16.4%	6.5%	13.7%	2.3%	20.9%	7.5%	11.9%	2.8%
	Speed between 0.5 to 1.5 m/s	38.4%	8.7%	31.7%	3.4%	43.1%	8.8%	41.5%	7.5%
	Speed between 1.5 to 2.5 m/s	23.6%	8.0%	23.7%	3.5%	20.1%	9.1%	30.8%	6.9%
	Speed above 2.5 m/s	10.4%	7.2%	17.5%	3.4%	5.9%	6.0%	6.2%	2.7%
**Rotational speed zones**	Abs. rot. speed betw. 0 to 25 °/s	34.3%	6.3%	31.5%	2.7%	39.1%	6.4%	29.3%	3.8%
	Abs. rot. speed betw. 25 to 50 °/s	27.5%	2.1%	26.9%	1.3%	29.1%	1.4%	25.4%	2.1%
	Abs. rot. speed betw. 50 to 100 °/s	23.4%	2.7%	24.1%	1.0%	21.7%	3.0%	25.6%	2.1%
	Abs. rot. speed above 100 °/s	14.7%	5.8%	17.5%	3.1%	10.0%	4.2%	19.5%	5.2%
	**“High-class”**	**Total (n = 39)**		**Basketball (n = 12)**		**Rugby (n = 12)**		**Tennis (n = 15)**	
		**Mean**	**Stdev.**	**Mean**	**Stdev.**	**Mean**	**Stdev.**	**Mean**	**Stdev.**
**WMP**	Avg. forward speed (m/s)	1.41	0.25	1.67	0.10	1.25	0.27	1.34	0.13
	Best forward speed (m/s)	2.52	0.54	3.14	0.16	2.32	0.58	2.17	0.14
	Avg. acceleration to 2 m (m/s^2^)	1.33	0.41	1.67	0.23	0.99	0.42	1.33	0.29
	Avg. rotational speed in a curve (°/s)	70	10	72	7	67	15	72	8
	Best rotational speed in a turn (°/s)	222	29	228	22	213	36	224	29
	Avg. rotational acceleration (°/s^2^)	262	128	422	119	181	33	199	26
**Max**	Maximal forward speed (m/s)	4.51	0.78	5.22	0.41	3.94	0.89	4.40	0.40
	Maximal rotational speed (°/s)	386	66	424	80	367	59	369	43
**Speed zones**	Speed <0 m/s	11.2%	3.3%	13.9%	3.1%	10.5%	1.7%	9.6%	3.2%
	Speed between 0 to 0.5 m/s	14.3%	4.8%	13.0%	2.3%	18.5%	6.0%	11.9%	2.8%
	Speed between 0.5 to 1.5 m/s	37.3%	8.6%	29.6%	3.4%	39.7%	8.7%	41.5%	7.5%
	Speed between 1.5 to 2.5 m/s	25.9%	7.4%	23.8%	3.8%	21.8%	7.6%	30.8%	6.9%
	Speed above 2.5 m/s	11.3%	7.3%	19.7%	3.2%	9.4%	7.1%	6.2%	2.7%
**Rotational speed zones**	Abs. rot. speed betw. 0 to 25 °/s	31.1%	4.1%	30.0%	2.5%	34.4%	4.1%	29.3%	3.8%
	Abs. rot. speed betw. 25 to 50 °/s	26.5%	2.0%	26.0%	0.9%	28.4%	1.4%	25.4%	2.1%
	Abs. rot. speed betw. 50 to 100 °/s	24.6%	2.2%	24.5%	1.1%	23.3%	2.7%	25.6%	2.1%
	Abs. rot. speed above 100 °/s	17.8%	4.6%	19.6%	2.1%	13.8%	2.7%	19.5%	5.2%

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
