# Peer review of "Wearable Wheelchair Mobility Performance Measurement in Basketball, Rugby, and Tennis: Lessons for Classification and Training"

_sensors, 2020, doi:10.3390/s20123518_

Round 1
Reviewer 1 Report
Please find my comments in the attached document.

Author Response
Thank you for the review, highly appreciated. Your comments and our response are included in the document attached.

Reviewer 2 Report
The reported research investigates whether wheelchair mobility peformance (WPB) is a critical performace factor in wheelchair basketball, wheelchair tennis, and wheelchair rugby and what aspects of WMP are fundamental for each sport. The article also investigates how WMP can affect classification guidelines and wheelchair design. A hypothesis is proposed that wearble technologies provide added value to wheelchair sports both in terms of performance and training. The limitations of this investigation are clearly reported. The outcomes can be used to optimize athlete training and wheelchair design for specific sports.
While I am not an expert in Paralympic sports, I really enjoyed reading this article. It should definitely be published afther the authors address minor editorial comments (see below). I was especially impressed by the fact that this study received no external funding. This shows true dedidcation and love of scientific research! As a researcher who is active both in reviewing and publishing I am beginning to notice a curious trend: studies with no external or little external funding often report much more interesting, insightful, and genuine results than studies that receive a great deal of funding.
My specific editorial comments are as follows.
1. I would recommend changing the title of your article to "Wearable Wheelchair Mobility Performance Measurement in Basketball, Rubby, and Tennis: Lessons for Classification, Training, and Wheechair Design"
This title, in my opinion, would reflect much better on the content of your article, the reported outcomes, and long-term implications.
2. line 34: remove "based" after "scientific".
3. line 49: "to what extend" --> "to what extent".
4. lines 56--57: "Wheelchair mobility performance closely relates to the athlete-wheelchair interaction, covering two of the main components in wheeled sports [6,7,8]."
What are the main components in wheeled sports? Do you mean the athlete and the wheelchair?
5. Line 70: "only athletes with irreversible upper extremity impairments are eligible"
Could you briefly explain what are "irreversible upper extremity impairments"? I don't know much about rubgy, but from what I've read about it, it appears to be an extreme contact sport similar to American football and ice hockey. One would naturally expect that wheelchair bound athletes must be really fit to play it. Your sentence, if I understand it correctly, implies that the athletes are eligible only if they have irreversible upper extermity impairments. Please explain it briefly for readers who lack background in Paralymic sports.
6. lines 84-86: "Yet, if WMP across sports of similarly impairment athletes show much resemblance, that might give cause to improve knowledge exchange between sports rather than sheer focus on the differences."
Rewrite this sentence. It is not grammatically correct. "between" --> "among", because you're talking about more than 2 sports; "sheer focus" is a noun that should not be juxtaposed to a verb "to improve..."
7. line 97: What is "elite level"? Could you briefly explain it to the general reader?
8. Table 1. How are the group numbers (1-2.5; 0.2-2; 3+, etc.) assigned/computed? Is there a standard classification? If yes, you should reference it.
9. lines 120 - 122: Please give references (e.g., urls) for the actual sensors for informed readers, if they so wish, to look up the technical characteristics of the sensors.
10. lines 148 - 149: Is it standard practice to give rotational velocities in degrees per second vs. radians per second.
11. lines 150 - 154: This paragraph seems to suggest that your analysis, at least partially, is based on measuring more compatible athletes. You should mention it in the abstract to alert the reader ahead of time.
12. I found Figures 2 and 3 somewhat confusing. It took me a while to comprehend different lines and colors. Tables with a brief explanation may be better for legibility.
13. Lines 205 -- 206: "only the average starting acceleration still showing significantly different."
This is not a grammatically correct phrase. Did you mean "significant differences"?
14. line 215: "motions will be more fluently"
"fluently" --> "fluent";
15. lines 225-234.
I enjoyed reading this insightful and honest discussion of the limitations of the reported study and how these limitations can be addressed in the future.
16. line 238: "extend" --> "extent"
17. line 257: Are the "high point players" the playes who score a lot?
18. Line 262: "formerly research" --> "prior research" or "previous research".
Author Response

(The authors gave the same response as above.)

Reviewer 3 Report
The paper studies WMP for three types of sports. And the results can be useful for direct classification and training guidelines.
Regarding the current manuscript, I have the following concerns:
1. What roles do the three sensors in wheelchairs? And how can they be used to compute speed? and was the data from the three sensors fused?
2. When evaluating the wheelchair mobility performance, does the study consider the weights and sizes of the athletes?
3. Is it possible to compare the study with some existing studies to show the advantages of the manuscript?
Author Response

(The authors gave the same response as above.)
